# Bimodal distribution of size-resolved particle effective density: results from a short campaign in a rural environment in the North China Plain

Yaqing Zhou[1,2], Nan Ma[1,2], Qiaoqiao Wang[1,2], Zhibin Wang[3], Chunrong Chen[4], Jiangchuan Tao[1,2], Juan Hong[1,2], Long Peng[1,2],

Yao He[1,2], Linhong Xie[1,2], Shaowen Zhu[1,2], Yuxuan Zhang[5,6], Guo Li[5], Wanyun Xu[7], Peng Cheng[8], Uwe Kuhn[5], Guangsheng Zhou[7], Pingqing Fu[9], Qiang Zhang[4], Hang Su[5,10], Yafang Cheng[5]

[1]Institute for Environmental and Climate Research, Jinan University, Guangzhou, 511443, China
[2]Guangdong-Hongkong-Macau Joint Laboratory of Collaborative Innovation for Environmental Quality, Guangzhou, 511443, China
[3]Research Center for Air Pollution and Health, College of Environmental and Resource Sciences, Zhejiang University, Hangzhou, 310058, China
[4]Department of Earth System Science, Tsinghua University, Beijing, 100084, China
[5]Multiphase Chemistry Department, Max Planck Institute for Chemistry, Mainz, 55128, Germany
[6]School of Atmospheric Sciences, Nanjing University, Nanjing, 210093, China
[7]State Key Laboratory of Severe Weather, Key Laboratory for Atmospheric Chemistry, Institute of Atmospheric Composition and Environmental Meteorology, Chinese Academy of Meteorological Sciences, Beijing, 100081, China
[8]Institute of Mass Spectrometry and Atmospheric Environment, Jinan University, Guangzhou, 510632, China
[9]Institute of Surface-Earth System Science, Tianjin University, Tianjin, 300072, China
[10]State Environmental Protection Key Laboratory of Formation and Prevention of Urban Air Pollution Complex, Shanghai Academy of Environmental Sciences, Shanghai, 200233, China

*Correspondence to*: Nan Ma (nan.ma@jnu.edu.cn) and Qiaoqiao Wang (q.wang2@outlook.com)

**Abstract.** Effective density is one of the most important physical properties of atmospheric particles. It is closely linked to particle chemical composition and morphology, and could provide special information on particle emissions and aging processes. In this study, size-resolved particle effective density was measured with a combined DMA-CPMA-CPC system in autumn 2019 as part of the Multiphase chemistry experiment in Fogs and Aerosols in the North China Plain (McFAN). With a new developed flexible Gaussian fit algorithm, frequent (77-87%) bimodal distribution of particle effective density is identified, with a low-density mode (named sub-density mode) accounting for 22-27% of total number of observed particles. The prevalence of the sub-density mode is closely related to fresh black carbon (BC) emissions. The geometric mean for the main-density mode ($\bar{\rho}_{\text{eff,main}}$) increases from 1.18±0.10 g cm$^{-3}$ (50 nm) to 1.37±0.12 g cm$^{-3}$ (300 nm) due to larger fraction of high-density components and more significant restructuring effect at large particle sizes, but decreases from 0.89±0.08 g cm$^{-3}$ (50 nm) to 0.62±0.12 g cm$^{-3}$ (300 nm) for the sub-density mode ($\bar{\rho}_{\text{eff,sub}}$) which could be mainly ascribed to the agglomerate effect of BC. $\bar{\rho}_{\text{eff,main}}$ and $\bar{\rho}_{\text{eff,sub}}$ show similar diurnal cycles with peaks in the early afternoon, mainly attributed to the increasing mass fraction of high material density components associated with secondary aerosol production, especially of secondary inorganic aerosol (SIA). To investigate the impact of chemical composition, bulk particle effective density was calculated based on measured chemical composition ($\rho_{\text{eff,ACSM}}$) and compared with the average effective density at 300 nm

($\bar{\rho}_{\text{eff,tot,300nm}}$). The best agreement between the two densities is achieved when assuming a BC effective density of 0.60 g cm$^{-3}$. The particle effective density is highly dependent on SIA and BC mass fractions. The influence of BC on the effective density is even stronger than SIA, implying the importance and necessity of including BC in the estimate of effective density for ambient particles.

## 1 Introduction

Atmospheric aerosol has a significant impact on air quality, climate change and public health (Dockery and Pope, 1994; IPCC, 2007; Laden et al., 2000; Su et al., 2020). This is determined by a combination of various particle physical and chemical properties. Density, as one of important physical properties of aerosol particles, is intimately associated with particles' optical, chemical and dynamic properties (Ditas et al., 2018; Nosko and Olofsson, 2017; Park et al., 2003). It serves as a link between particle mass and mobility size (McMurry et al., 2002; Schmid et al., 2007), and is usually used to infer particle morphology,

chemical composition, and associated atmospheric processes such as emission and aging (Abegglen et al., 2015; Levy et al., 2013; Olfert et al., 2007; Olfert et al., 2017; Park et al., 2003; Wang et al., 2018; Wu et al., 2019; Zhang et al., 2018). Owing to the inability to directly measure the density of atmospheric particles, effective density is usually applied practically in aerosol research.

   The effective density is closely linked to particle chemical composition and morphology. It is generally observed within the

range of 1.06-1.81 g cm$^{-3}$ in the ambient atmosphere (Cha and Olofsson, 2018; Geller et al., 2006; Hu et al., 2012; Levy et al., 2014; Lin et al., 2018; Rissler et al., 2014; Zamora et al., 2019), of which high values are attributed to the dominant of ammonium sulfate ($(NH_4)_2SO_4$), ammonium nitrate ($NH_4NO_3$) and metals (Zhai et al., 2017; Zhang et al., 2016a). In contrast, the effective density of black carbon (BC) or organic aerosol (OA) dominated particles is relatively low (Zhai et al., 2017). While the effective densities of inorganic components are well recognized, there exists large uncertainties in the effective

densities of both BC and OA (Li et al., 2016; Malloy et al., 2009; Zhang et al., 2008), leading to significant variations in the effective densities of ambient bulk particles.

   The variability of OA density primarily originates from the diversity of organic species, formation mechanisms and aging processes. For instance, the effective density of secondary organic aerosol (SOA) oxidized from *m*-xylene, terpenes and cycloalkenes was estimated to be around 1.10 g cm$^{-3}$ for those generated with inorganic seeds but around 1.40 g cm$^{-3}$ without

seeds (Bahreini et al., 2005). Malloy et al., (2009) determined effective densities of 1.24 g cm$^{-3}$ and 1.35 g cm$^{-3}$ for SOA from ozonolysis of *α*-pinene and photo-oxidation of *m*-xylene, respectively. An increase of 10% in effective density caused by oxidative aging process was also observed for SOA (George and Abbatt, 2010). Little information is found for effective density of primary organic aerosol (POA), but its value is usually lower than that of SOA due to closely related to fresh emissions (Nakao et al., 2011). Generally, a simplified average value around 1.20-1.40 g cm$^{-3}$ is commonly applied for ambient OA in

numerous studies (e.g. Hallquist et al., 2009; Hu et al., 2012; Levy et al., 2013).

A more dramatic variation is observed for the effective density of BC, ranging from 0.10 to 1.80 g cm$^{-3}$ (Pagels et al., 2009; Peng et al., 2016). The lower end of the range is generally found for freshly emitted BC, owing to the presence of non-spherical morphology. For example, Zhang et al., (2008) found that soot particles generated under incomplete combustion condition could have an effective density as low as 0.10 and 0.56 g cm$^{-3}$ for particle size at 360 nm and 50 nm, respectively. Soot particles emitted from car brake materials, representing a typical traffic-relate source, were also found to be at a low effective density of 0.75 g cm$^{-3}$ (Nakao et al., 2011; Nosko and Olofsson, 2017). The effective density increases immediately with aging, as a more compact morphology of BC is formed (Ghazi and Olfert, 2013; Pagels et al., 2009). For instance, a 3-10 folds increment was found for the effective density of coated soot particles compared to fresh ones, attributed to the morphology compaction verified by transmission electron microscopy (TEM) technique (Zhang et al., 2008).

Currently, there exists several techniques for the measurement of particle effective density. One commonly used method is measure particle mobility size and mass simultaneously based on differential mobility analyzer (DMA) coupled with centrifugal particle mass analyzer (CPMA) or aerosol particle mass analyzer (APM) (Geller et al., 2006; McMurry et al., 2002; Rissler et al., 2014; Wang et al., 2018), and calculate the effective density assuming a spherical morphology. Another alternative technique is based on the relationship between aerodynamic diameter and mobility diameter, measured by scanning mobility particle sizer-electrical low-pressure impactor (SMPS-ELPI) or DMA-micro-orifice uniform deposit impactor (DMA-MOUDI) (Kelly and McMurry, 1992; Nosko and Olofsson, 2017). Techniques such as connecting a mass spectrometer in series with a DMA are also applied in field studies as well as laboratory experiments, which can directly provide particle chemical composition information in addition to effective density (Spencer et al., 2007; Zhang et al., 2016a). Among the above approaches, the first one (mass-volume based method) is the most widely used in ambient measurements (Peng et al., 2021).

Measured effective densities of ambient particles with a certain diameter usually present as a broad frequency distribution. To quantify the average and diversity of the measured effective densities, a Gaussian-based unimodal fit is commonly used. However, a bimodal effective density distribution with a second density mode occurring at density below 1.00 g cm$^{-3}$ are sometimes observed when the air is strongly affected by primary emissions (Levy et al., 2013; Ma et al., 2020). The contribution of the second density mode can reach as high as 92% (Rissler et al., 2014). It was suggested that the second density mode is dominated by soot and can be used to infer the contribution of fresh emission as well as the morphology of freshly emitted particles (Kuwata et al., 2009). To date, only a few works performed a bimodal fit on measured effective density distribution, resulting in scarce information of the second density mode for ambient particles. In addition, most studies reported values of effective density by taking the peak of the distribution (Ma et al., 2017; Qiao et al., 2018), which hides the information of variability of the density distribution.

In this study, size-resolved effective density of ambient particles was measured with a DMA-CPMA-CPC system during Multiphase chemistry experiment in Fogs and Aerosols in the North China Plain (McFAN) in Hebei province in October-November, 2019. A new developed flexible Gaussian fit approach is applied to separate two different density modes, and a comprehensive information of each density mode is given to fully characterize variations of effective density for ambient

particles. Furthermore, measured effective density is compared with the calculated one to investigate the connections between
particle effective density and chemical composition.

## 2 Experiments

### 2.1 Sampling site and instrumentation

Observation of particle effective density, chemical composition and number size distribution were performed from 18 October to 1 November of 2019 at Gucheng (39°09' N, 115°44' E) inside the Ecological and Agricultural Meteorological Test
Station of the Chinese Academy of Meteorological Sciences in Hebei province in China. This site is located to the southwest of Beijing (~100 km) and northeast of Baoding, Hebei province (~35 km). As shown in Fig. S1, the green area in the right panel is farmland and the faint yellow areas are scattered villages. The sampling site (red circle in Fig. S1) is surrounded by agricultural fields (mainly for corn cultivation) and is ~ 1.5 km away from the No.107 National Way. There is no significant emission sources such as large factories and power plants within 20 km. The main anthropogenic sources are biomass and coal
combustion for domestic heating and cooking, as well as traffic in the country roads connecting villages. It should be noted that the heating season in China normally starts from 15 November to 15 March of the following year, which is not covered by our sampling period. Besides, the temperature during the measurement period varies from 0 to 25 °C (Fig. S2). The emission from heating is therefore considered to be limited. Moreover, source apportionment results also imply that the sampling site is not significantly affected by coal combustion during the observation period (Fig. S2). As shown in Fig. S1, one of the prevailing
winds at the site is from southwest with relatively high wind speed, possibly indicating the influence of regional transport from southern Hebei province. Overall, the sampling site is influenced by both local emissions from nearby villages and regional transport, and can well represent the average pollution condition of the rural area in the North China Plain (NCP) (Li et al., 2021a).

During the observation, all aerosol instruments were settled inside an air-conditioned container with a constant temperature
around 24 °C. The schematic of our measurement settings is shown in Fig. S3. Ambient particles were sampled with a $PM_{10}$ inlet and then passed through a 1.2 m long Nafion dryer and a 0.4 m long silica gel drying tube in series to reduce the relative humidity (RH) to < 30%. A constant flowrate of 16.7 L $min^{-1}$ in the main inlet line were maintained by a mass flow controller and an extra pump to ensure an accurate cut-off size of the inlet. There could be some possible influences from the pre-sampling treatment of ambient aerosols. One is the drying process in the sampling line since the evaporation of particle water content
may affect its morphology. To date, only a few studies have investigated the influence of drying process on the particle effective density. Pagels et al. (2009) found that the influence of drying from RH of 80% to 5% could be negligible for 150 nm soot particles coated with $(NH_4)_2SO_4$. Yuan et al. (2020) concluded that the effective density of 240 nm BC particles coated with $NH_4NO_3$ respectively decreased by 5% and 16% for thickly and thinly coated particles when dry the particles from RH of 70% to 5%. According to these studies, the influence of the drying process on our effective density measurement is assumed

to be negligible. Another possible influencing factor is the particle losses in the sampling line. The measurements were conducted for the aerosol particles in the size range from 50 to 500 nm, which are not very sensitive to the three main loss mechanisms (i.e., diffusion loss, sedimentation loss and impaction loss) (Baron and Willeke, 2001; Von Der Weiden et al., 2009). Furthermore, the particle losses mainly affect the absolute particle number concentration. Its influence on the measurement of particle effective density is therefore considered to be negligible.

A combined DMA-CPMA-CPC system was employed to measure the size-resolved effective density of particles with mobility diameter of 50, 100, 150, 220, and 300 nm. Aerosol particles firstly passed through a DMA (Model 3081, TSI Inc.) operated at a sample flowrate of 1.0 L min$^{-1}$ and a sheath flowrate of 5.0 L min$^{-1}$ for size classification. The output semi-monodisperse charged particles were then introduced into a CPMA (Cambustion Ltd.) followed by a CPC (Model 3772, TSI Inc.) to measure their mass distribution. It takes about one hour for a complete measurement cycle, in which frequency

distributions of effective density with 22-38 bins for the 5 selected diameters were obtained. The measurement uncertainty of the DMA-CPMA-CPC system could come from two aspects: the size classification of DMA and the mass classification of CPMA. Based on the test using polystyrene latex (PSL) particles with diameters of 150, 220, and 300 nm, an average sizing uncertainty of ±2% was determined for our DMA (Fig. S4). This uncertainty is similar to the value of ±1% (±1 nm) reported by Mulholland et al. (1999) for the size range of 100-300 nm at an aerosol-sheath flow rate ratio of 0.1. The uncertainty of the

mass classification of CPMA is estimated as 1.4% according to the results of Taylor and Kuyatt (1994) and Symonds et al. (2013). The overall measurement uncertainty of the DMA-CPMA-CPC system were also evaluated using PSL particles with diameter of 150, 220, and 300 nm before and after the field campaign. An uncertainty within ~ 5% was found by comparing the measured effective densities with PSL material density (1.05 g cm$^{-3}$) (Fig. S5).

    Particle size distribution between 13-700 nm was measured by scanning mobility particle sizer (SMPS, Model 3938, TSI

Inc.). The mass concentrations of non-refractory submicron aerosol (NR-PM$_1$) composition including OA, nitrate (NO$_3^-$), sulfate (SO$_4^{2-}$), ammonium (NH$_4^+$) and chloride (Cl$^-$) were acquired using a quadrupole aerosol chemical speciation monitor (Q-ACSM, Aerodyne Research Inc.) with a time-resolution of 30 min. The setup and calibration procedures followed that described in Ng et al., (2011). Source apportionment for OA was conducted using positive matrix factorization (PMF) algorithm, and four factors including one SOA (oxidized OA, OOA) and three POA, i.e., biomass burning OA (BBOA),

cooking OA (COA) and hydrocarbon-like OA (HOA) were determined. BC mass concentration was measured with a 7-wavelength Aethalometer (Model AE33, Magee Scientific) based on the attenuation at $\lambda$ = 880 nm and a corresponding mass absorption cross section of 7.77 m$^2$ g$^{-1}$ (Drinovec et al., 2015).

    It should be noted that the effective density and size distribution data presented in our work starts from 18 October to 1 November, while chemical data are available between 18 October and 27 October. Effective density characterization in Sect.

3.1-3.4 is based on the data covering the entire observation period, and the combined analysis of density and chemical composition (Sect. 3.5) only covers the shorter period mentioned above.

## 2.2 Calculation of effective density

The mass-volume based method was applied in our study. Particle effective density can be calculated as:

$$\rho_{\text{eff}} = \frac{m}{\frac{\pi}{6}D_{\text{me}}^3} \tag{1}$$

where $m$ is the particle mass measured by CPMA, $D_{\text{me}}$ is the particle mobility size measured by DMA. It can be speculated that $\rho_{\text{eff}}$ is equal to the material density for compact spherical particles, while $\rho_{\text{eff}}$ is smaller than the material density for non-spherical, irregular particles or spherical but porous particles.

In contrast to previous studies mostly using unimodal Gaussian fit to process raw data of DMA-CPMA-CPC measurements, a flexible Gaussian fit algorithm with either uni- or bimodal combined with corresponding classification standard was developed in our study to better characterize the frequency distribution of particle effective density. The algorithm includes two steps.

The first step is to remove doubly charged particles. It is shown that doubly charged particles is ubiquitous and unavoidable for DMA based mobility-selected method, especially for particle size over 100 nm (Park et al., 2003). Nonetheless, doubly charged particles can be cleanly separated from singly charged particles with the same mobility size after penetrating through CPMA (McMurry et al., 2002). The theoretical masses of doubly charged particles for the five selected mobility sizes were estimated based on the Boltzmann equilibrium charge distribution (Fuchs, 1963; Wiedensohler, 1988) and listed in Table S1. For instance, doubly charged particles present a 2.8-fold higher mass value than the singly charged particles with DMA selected size of 300 nm. For each acquired particle mass distribution, the mode of doubly charged particles were discarded for the following analysis. Note that other multiply charged particles are not considered here since their mass is beyond the CPMA mass setting range.

The second step is to determine the number of modes and perform a uni- or bimodal Gaussian fit. For each effective density distribution, a bimodal Gaussian fit (Eq. 2 with $N_{\text{mode}} = 2$) was firstly implemented. If the height $a_i$ of the mode located at lower density mode exceeds 5% of that of the other mode, the fitting terminated. Otherwise, a unimodal Gaussian fit (Eq. 2 with $N_{\text{mode}} = 1$) was performed.

$$f(\rho_{\text{bin}}) = \sum_{i=1}^{N_{\text{mode}}} a_i \cdot \exp\left(\frac{\left(\log\rho_{\text{bin}} - \log\bar{\rho}_{\text{eff,i}}\right)^2}{2\sigma_i^2}\right) \tag{2}$$

where, $f$ is the frequency distribution of effective density; $\rho_{\text{bin}}$ is the effective density of each measured bin and is equally spaced in logarithm scale; $N_{\text{mode}}$ is the number of modes used in the fit; $\bar{\rho}_{\text{eff,i}}$ is the geometric mean effective density of mode $i$; $a_i$ is the height at $\bar{\rho}_{\text{eff,i}}$ of mode $i$; $\sigma_i$ is the standard deviation of mode $i$. The uncertainty of each individual Gaussian fit could be estimated based on the variation of the fitted $\bar{\rho}_{\text{eff,i}}$ in each mode at the 95% confidence level. And the overall uncertainty is estimated by averaging the uncertainties of all the fits, which gives averages within 2.5% and 7.0% for $\bar{\rho}_{\text{eff,main}}$ and $\bar{\rho}_{\text{eff,sub}}$ at the five measured sizes, respectively. This uncertainty range is similar to the measurement uncertainty discussed in Sect. 2.1.

The bulk particle effective density ($\rho_{\text{eff,ACSM}}$) was also calculated with the aerosol chemical composition measured by ACSM based on Eq. 3, assuming that ambient particles were composed of $(NH_4)_2SO_4$, $NH_4NO_3$, ammonium chloride ($NH_4Cl$), OA and BC.

$$\frac{1}{\rho_{\text{eff,ACSM}}} = \sum_{i=1}^{N} \frac{f_{\text{m,i}}}{\rho_{\text{i}}} \tag{3}$$

where $f_{\text{m,i}}$ represents the mass fraction of chemical component $i$; $\rho_{\text{i}}$ is the assumed effective density of chemical component $i$. Densities of $(NH_4)_2SO_4$, $NH_4NO_3$, $NH_4Cl$ are 1.76, 1.73 and 1.53 g cm$^{-3}$, respectively. The density of OA usually ranges from 1.20 to 1.60 g cm$^{-3}$ (Dinar et al., 2006; Kostenidou et al., 2007; Turpin and Lim, 2001). In this study a fixed value of 1.30 g cm$^{-3}$ was used in the calculation. A sensitivity test regarding the uncertainties associated with the range of OA density was conducted. Considering the large range of observed BC effective density (0.30-1.80 g cm$^{-3}$), a sensitivity analysis was applied to identify the optimal value of BC effective density. More discussion on OA and BC effective density can be found in Sect. 3.5.

For the comparison with $\rho_{\text{eff,ACSM}}$, we also estimated the average effective density based on the measured frequency distribution of effective density as:

$$\bar{\rho}_{\text{eff,tot}} = \frac{\int \rho_{\text{bin}} \cdot \frac{dn}{d\log\rho_{\text{bin}}} d\log\rho_{\text{bin}}}{\int \frac{dn}{d\log\rho_{\text{bin}}} d\log\rho_{\text{bin}}} \tag{4}$$

where, $\frac{dn}{d\log\rho_{\text{bin}}}$ is the measured frequency distribution of effective density.

**2.3 Calculation of fractal dimension**

With the identification of sub-density mode in Sect 2.2, the geometric mean of the measured mass distribution of the sub-density mode ($\bar{m}_{\text{sub}}$) can be used to characterize the mass-mobility relationship in the size range of 50-300 nm. The relationship could be expressed as (Park et al., 2003)

$$\bar{m}_{\text{sub}} = C d_{\text{me}}^{D_{\text{f}}} \tag{5}$$

where $C$ is a constant and $D_{\text{f}}$ is the fractal dimension, which reflects the morphology of particles. $D_{\text{f}}$ varies from 1 to 3, increasing as particle morphology becomes less fractal and with the value of 3 for spherical particle. By fitting $\bar{m}_{\text{sub}}$ and $d_{\text{me}}$ at the five measured particle sizes, $C$ and $D_{\text{f}}$ can be determined. The measurement cycles containing less than 4 sizes with identified sub-density mode were excluded in the calculation to ensure the accuracy.

**3 Results and discussion**

**3.1 Size-resolved particle effective densities**

Figure 1 depicts the statistical results after averaging all effective density distributions for 50, 100, 150, 220 and 300 nm particle sizes. Based on the flexible Gaussian fit, a considerable amount (77-87%) of bimodal distribution condition occurred

for all size particles. The first peak is found to be located below 0.90 g cm$^{-3}$, and the second is higher than 1.20 g cm$^{-3}$. For a compositional mixed compact spherical particle, particle density speculated from the material density of each individual component is supposed to be higher than 1.20 g cm$^{-3}$ (the lowest material density among all components is 1.20 g cm$^{-3}$, Ma et al., 2017; Xie et al., 2017). However, a low-density mode with peak density less than 1.00 g cm$^{-3}$ usually exists for freshly emitted particles, indicating the presence of particles with highly agglomerated or porous morphology (Park et al., 2003; Rissler

et al., 2013). Therefore, the two density modes observed in this study are defined as: (1) "sub-density" referred to low effective density mode, implying particles' feature of fractal aggregated; (2) "main-density" referred to relatively high effective density mode with a dense or compact particle structure.

    As listed in Table 1, the occurrence frequencies of the sub-density mode are 79%, 87%, 86%, 77%, and 81% for particle diameters of 50, 100, 150, 220, and 300 nm, respectively. These frequencies are significantly higher than those reported in

Levy et al., (2013) which only reach up to 32% for 150 nm particles in an inner-city in Houston. It is also different from another study conducted in suburban environment of Nanjing, China (Ma et al., 2017), where they observed a dominant unimodal distribution of effective density with only < 10% occurrence with bimodal distribution. On the other hand, a low effective density mode (density < 1.20 g cm$^{-3}$) exists or even dominates in the measurements near emission sources, which is ascribed to freshly emitted particles and non-uniformly mixed particles (Nosko and Olofsson, 2017; Olfert and Rogak, 2019;

Park et al., 2003). Numerous studies have found low effective densities of freshly emitted BC, with a minimum of 0.10 g cm$^{-3}$ (Pagels et al., 2009). While the density of OA is usually assumed as 1.2-1.3 g cm$^{-3}$ in most cases (Hallquist et al., 2009), some studies have found it could be as low as 0.6-1.1 g cm$^{-3}$ (Nakao et al., 2011; Li et al., 2016). To eluciate the role of these two components in the sub-density mode, we further analyze the correlation between the number fraction of the sub-density mode ($F_{sub}$) and the mass fractions of BC and OA. As seen in Fig. S6-S7, the mass fraction of BC shows significant correlation

with $F_{sub}$ at 150, 220 and 300 nm ($R^2 = 0.46$-$0.57$), whereas barely no correlation is observed between OA and $F_{sub}$ ($R^2 = 0.02$-$0.09$), implying that the sub-density mode at these three sizes could be mainly attributed to freshly emitted BC and the quantity of the sub-density mode is closely related to the variation of BC mass fraction. However, $F_{sub}$ at 50-100 nm shows little correlation with either BC or OA mass fraction, which could be explained by the difference between the PM$_1$ bulk chemical composition and the chemical composition of particles smaller than 100 nm. In addition, the discrepancy in observed

frequencies of the sub-density mode with other studies can be explained, to some extent, by different thresholds used in the studies to identify bimodal distribution of effective density. In our study, we aim at identifying as more bimodal cases as possible with a low threshold of 5% as mentioned in Sect. 2.2.

    The mean (min., max.) values of $F_{sub}$ are 27% (4%, 88%), 22% (5%, 63%), 23% (6%, 64%), 23% (9%, 85%), 25% (5%, 78%) for 50, 100, 150, 220 and 300 nm particle size, respectively (Table 1), showing no obvious trend with particle size. The

results in our study are lower than those (11-92%, with most cases exceeding 50%) in traffic emission influenced sites (Geller et al., 2006; Rissler et al., 2014), as the number fraction of the sub-density mode is tightly related to the influence of fresh BC emission sources (Geller et al., 2006; Levy et al., 2014; Ma et al., 2020). Interestingly, when we classify the measured particle

number size distribution according to the measured $F_{sub}$ and use the 25th and 75th percentiles of $F_{sub}$ at each measured particle size as threshold, we found a more prominent Aitken mode with higher $F_{sub}$ ($F_{sub}$ > 75th $F_{sub}$) (Fig. S8). The initial burst of Aitken mode particles may be attributable to the enhanced traffic related emissions (Xie et al., 2017). Previous studies showed that the effective density of 50 nm traffic-emitted particles could be below 1.0 g cm$^{-3}$ (Olfert et al., 2007; Park et al., 2003; Momenimovahed and Olfert, 2015). Therefore, the observed higher Aitken mode in our study may stem from the higher contribution of traffic emission, and subsequently lead to an increase of particles in the sub-density mode. This finding also provides a good support for the connections between $F_{sub}$ and fresh emission sources.

During the entire campaign, the geometric mean for the main-density mode ($\bar{\rho}_{eff,main}$) vary from 0.94 to 1.42, 1.01 to 1.47, 1.09 to 1.48, 1.16 to 1.59, 1.22 to 1.73 g cm$^{-3}$ for particles at 50, 100, 150, 220 and 300 nm respectively, and for the sub-density mode ($\bar{\rho}_{eff,sub}$) vary from 0.74 to 1.20, 0.59 to 1.25, 0.55 to 1.14, 0.36 to 0.99, 0.34 to 1.00 g cm$^{-3}$, respectively. Although the temporal variations are similar for the five measured sizes, densities of particles at 50 and 100 nm show greater variability than larger ones (Fig. S9).

$\bar{\rho}_{eff,main}$ exhibits an evident ascending trend with particle size (Fig. 2), similar to the pattern reported in Shanghai (Yin et al., 2015). This is probably affected by two aspects. One is the increase of the proportion of high-density components with increasing particle size, from OA dominant to secondary inorganic aerosol (SIA) dominant. Although size-resolved chemical composition was not measured in this study, it can be inferred from Li et al., (2021b) that the mass fraction of SIA increased from 49% (49%) to 55% (61%) with particle size increasing from 150 to 300 nm in autumn (winter) in the NCP. The other is the collapse and restructuring effect occurred for soot-containing particles during aging processes (Ghazi and Olfert, 2013; Pagels et al., 2009), leading to a more spherical-like or more dense morphology of large particles.

However, $\bar{\rho}_{eff,sub}$ decreases from 0.89 g cm$^{-3}$ to 0.62 g cm$^{-3}$ as particle size increases from 50 to 300 nm. This can be ascribed to the agglomerate effect that fresh soot particles with larger sizes are more agglomerated and hence have a larger fractal dimension and lower effective density. This has been confirmed by TEM images for diesel engine exhaust particles (Park et al., 2003) and traffic-related environmental samples (Rissler et al., 2014). Figure 2 also shows the results for fresh soot from previous studies. All the reported densities of freshly emitted particles from diesel combustion, aircraft, and other burning sources show decreasing trends with particle size, but with much steeper slopes than our results. This is because after emitted into the atmosphere, fresh soot is rapidly coated with $(NH_4)_2SO_4$, $NH_4NO_3$ and OA (Kuwata et al., 2009), leading to an increasing $\bar{\rho}_{eff,sub}$ and less pronounced decline trend compared with observations for primary sources (Olfert et al., 2007; Tavakoli and Olfert, 2014; Ubogu et al., 2018).

Most previous studies did not differentiate the two modes and reported either an overall increasing (Levy et al., 2014; Ma et al., 2020; Yin et al., 2015) or decreasing trends (Geller et al., 2006; Qiao et al., 2018) of $\bar{\rho}_{eff}$ with particle size. A unimodal distribution implies an assumption of an internal mixing of particles, which is not true since most surface atmosphere is affected by fresh emissions. Therefore, the feature of the sub-density mode may be missed if a unimodal fit is used. As shown in Fig. 2, only a slightly monotonically increasing trend of the average effective density ($\bar{\rho}_{eff,tot}$) with particle size are observed (grey

dotted line in Fig. 2). Therefore, sub-density modes need to be isolated to reflect the detailed features of effective density of ambient particles. Furthermore, bimodal fits can also provide additional information on the mixing state of fresh and aged particles.

In addition to the size dependence of $\bar{\rho}_{eff,main}$ and $\bar{\rho}_{eff,sub}$, $\sigma$ of different effective density modes also varies with particle size. The mean $\sigma$ of the main-density mode are similar for particles in the size range of 50 to 150 nm (~0.053), and gradually increases to 0.067 at 300 nm (Table 1). The mean $\sigma$ of the sub-density mode demonstrates the highest value (~0.215) at 300 nm, and decreases to ~0.044 at 50 nm. Since particles usually appear as more dense morphology in the main-density mode (Rissler et al., 2014), the variation of $\sigma$ probably originates from the differences in particle chemical composition. For the sub-density mode, soot is the main component of particles (approximately over 80%, Kuwata et al., 2009). It has been found that the morphology of freshly emitted soot is more chain-like at large particle sizes than at small sizes (e.g., Park et al., 2003). We therefore speculate that the chain-like morphology enables a higher degree of morphological variation of larger particles, resulting in a broader sub-density distribution.

## 3.2 Evolution of effective density with meteorology conditions

Weather conditions play a crucial role in the formation, aging and emission transportation of aerosol, and may therefore affect the distribution, composition, mixing state and consequently also the effective density of ambient aerosol. As seen in Fig. S2, the pollution level at the sampling site is sensitive to the variations of wind speed and direction during the observation period. Low $PM_{0.7}$ concentrations were usually presented with strong northerly winds while high $PM_{0.7}$ concentrations were associated with calm winds or southwest winds.

Figure 3 and Fig. S10 shows the average $\bar{\rho}_{eff,main}$, $\bar{\rho}_{eff,sub}$ and $F_{sub}$ at each specific wind speed and direction. Obvious high values of $\bar{\rho}_{eff,main}$ and $\bar{\rho}_{eff,sub}$ appear with wind direction of southwest and wind speed > 2 m s$^{-1}$. This pattern clearly indicates the influence of regional transport from southern Hebei province, an area greatly affected by emissions from industrial and residential sources (Huang et al., 2019; Li et al., 2017). Air masses from this direction may bring pollutants with sufficient aging process, leading to changes in particle chemical composition and morphology, and consequently an increase in the fraction of particles closer to spherical with higher effective densities. Accordingly, $F_{sub}$ also shows distinctly low values (Fig. 3c and Fig. S10). It is worth mentioning that $\bar{\rho}_{eff,main}$ and $\bar{\rho}_{eff,sub}$ do not show any obvious difference for wind direction of northwest. This implies that the influence of the traffic emission at No.107 National Way, which is approximate 1.5 km away from the sampling site (Fig. S1), on our measurements is somehow limited. We also noticed that an increasing trend of $\bar{\rho}_{eff,main}$ is presented with increasing wind speed (Fig. S11). This increase could be interpreted by the antagonism between well-aged particles from long-range transport and fresh particles from local emissions. High wind speed is usually accompanied with the long-range transport of particles with sufficient aging and consequently high effective density; while low wind speed generally implies higher contribution of local fresh emissions, resulting in more particles with non-spherical morphology.

### 3.3 Effective density at different pollution levels

Given that the particle effective density varies dramatically with time (Fig. S9), the entire sampling period is classified into two pollution levels to elucidate the evolution of effective density under different pollution conditions. As mentioned in Sect. 2.1, particle mass concentration data only covers a part of the sampling period, and thus $PM_{0.7}$ volume concentration calculated based on SMPS measurement (size range of 13-700 nm) is applied to separate the sampling period into two groups: a more polluted group with $PM_{0.7}$ volume concentration higher than 50 $\mu m^3$ $cm^{-3}$, and a less polluted group corresponding to $PM_{0.7}$ volume concentration lower than 50 $\mu m^3$ $cm^{-3}$. It should be mentioned that the threshold of $PM_{0.7}$ volume concentration is comparable to $PM_1$ mass concentration of 60 $\mu g$ $m^{-3}$ ($R^2 = 0.97$, slope = 0.84, Fig. S12). The difference mainly stems from the size truncation of SMPS, as well as the time-dependent and size-dependent variations of the particle effective density (Morawska et al., 1999). Since particles are assumed to be spherical (i.e., fractal dimension = 3.0) in the calculation of $PM_{0.7}$, the obtained $PM_{0.7}$ may be overestimated.

The average effective density distributions of the two groups are depicted in Fig. 4 and the statistics of the $\bar{\rho}_{eff,main}$ and $\bar{\rho}_{eff,sub}$, as well as $F_{sub}$ and occurrence frequency of the sub-density mode are given in Table 2. $\bar{\rho}_{eff,main}$ in the more polluted group for each particle size is 0.08-0.11 g $cm^{-3}$ lower than that in the less polluted group. In contrast, the difference of $\bar{\rho}_{eff,sub}$ is insignificant with the maximum discrepancy of 0.03 g $cm^{-3}$, which is only one-third of the maximum discrepancy for $\bar{\rho}_{eff,main}$. The size dependence of $\bar{\rho}_{eff,main}$ and $\bar{\rho}_{eff,sub}$ follows the similar pattern under all conditions (Fig. S13).

It can be noted from Table 2 that $F_{sub}$ varies with pollution levels. In the less polluted group, $F_{sub}$ mainly distributes over a small range of 21-26%, with little size dependence; while in the more polluted group, $F_{sub}$ gradually decreases from 42% at 50 nm to 18% at 300 nm. This is consistent with the observations conducted by Rissler et al. (2014) in Copenhagen, which showed a decrease around 50% in the number fraction of fresh soot particles (equivalent to $F_{sub}$ in our study) from 75 nm to 350 nm under heavily polluted condition. Since the integrated area of the sub-density mode at each particle size are stable in the more polluted group, the size dependence of $F_{sub}$ is mainly driven by the increase in the integrated area of the main-density mode with particle size. As a result, $F_{sub}$ for particles at 50-100 nm is higher in the more polluted group than the less polluted group, but it is opposite for particles at 150-300 nm.

Similarly, the occurrence frequency of the sub-density mode decreases with particle size (with an exception of 50 nm) in the more polluted group: 72%, 95%, 89%, 63%, and 57% at 50, 100, 150, 220 and 300 nm, respectively. In contrast, the variation with particle size in the less polluted group is no more than 10%. One possible explanation is that the sub-density mode inherently diminishes due to higher aging degree for larger particles under the more polluted condition. Xie et al., (2017) found that the effective density distribution was more inclined to a unimodal distribution for particles at 220 nm than 40 nm during episodes at $PM_1$ volume concentrations above 50 $\mu m^3$ $cm^{-3}$. Hence, the occurrence probability of bimodal distribution was expected to be higher for smaller particles with respect to large particles, similar to the results in this study. Additionally, it is difficult to identify the sub-density mode as it falls below the threshold defined in Sect. 2.2 due to the increasing

contribution of the main-density mode for larger particles, which consequently reduces the corresponding occurrence frequency of the sub-density mode.

### 3.4 Diurnal variations of effective density

Figure 5 describes the diurnal cycle of $\bar{\rho}_{eff,main}$ and $\bar{\rho}_{eff,sub}$ at the five measured particles sizes. $\bar{\rho}_{eff,main}$ increases in the morning and gradually decreases in the late afternoon and early evening for all particle sizes. The upward tendency in the morning and early afternoon is associated with the increasing intensity of secondary aerosol production, resulting in an increment of high material density components, such as SIA and OOA mass fraction, especially of $NH_4NO_3$ (Fig. S14). Besides, with the increase in the fraction of internally mixed particles and coating thickness (Cheng et al., 2012; Ma et al., 2012), particles gradually approach spherical-like morphology (Ghazi and Olfert, 2013; Pagels et al., 2009), which also leads to the increase of $\bar{\rho}_{eff,main}$. In the late afternoon, $\bar{\rho}_{eff,main}$ starts to decline, attributed to elevated mass fractions of BC and POA (Fig. S14) due to enhanced emissions from traffic, biomass burning, and cooking identified by PMF.

The diurnal cycle of $\bar{\rho}_{eff,sub}$ is similar to that of $\bar{\rho}_{eff,main}$ for each particle size, with a daily maximum observed in the early afternoon and a relatively low level in the night. One difference between the two modes is that $\bar{\rho}_{eff,sub}$ drops more rapidly before sunset compared with $\bar{\rho}_{eff,main}$, especially for large particles ($\geq 150$ nm). This is because the sub-density mode is more directly related with freshly emitted particles than the main-density mode.

The diurnal cycle of $F_{sub}$ is different from the pattern of $\bar{\rho}_{eff,main}$ and $\bar{\rho}_{eff,sub}$. As shown in Fig. 5f, $F_{sub}$ exhibits insignificant daily variability for particles at 50-100 nm, while double peaks are observed in the morning and evening for particles at 150-300 nm associated with large amount of fresh BC emissions. The minimum $F_{sub}$ for particles at 150-300 nm is generally found during the mid-afternoon, e.g. 15% at 14:00 for 300 nm. This is consistent with the pattern of the occurrence frequency of the sub-density mode, which drops remarkably during 12:00-16:00 (Fig. S15). The drop of both $F_{sub}$ and occurrence frequency could be explained by the transition of particles from the sub-density mode to the main-density mode associated with aging processes.

As shown in Fig. 6, $D_f$ shows a distinct diurnal variation. There are two decreasing trends: from 06:00 to 11:00, and from 17:00 to 19:00, corresponding to the morning and evening rushing hours, with $D_f$ falling into the range of particles from vehicle emissions (2.22-2.84) reported in previous studies (Wu et al., 2019; Olfert et al., 2007; Park et al., 2003). The rapid increase in $D_f$ from 2.62 (11:00) to 2.86 (13:00) at noon could be attributed to photochemical aging, resulting in more compact particles (Ghazi and Olfert, 2013). During early afternoon (14:00-16:00), there is no $D_f$ values due to a sharp reduction in occurrence frequency of the sub-density mode (Fig. S15) as discussed above. $D_f$ gradually increases from 2.51 at 19:00 to 2.81 at 24:00, probably associated with aging processes during the night. It is worth noting that the increase rate of $D_f$ differs between noon and night, being ~0.12 h$^{-1}$ at noon which is twice of the night-time increase rate (0.06 h$^{-1}$). To minimize the influence of pollution level on the transiting particles from fractal to compact morphology, the variation of $D_f$ under the more polluted and the less polluted periods were further examined separately. As shown in Fig. S16, similar diurnal variations were observed at

two different pollution levels, with a higher $D_f$ increase rate at noon than night-time (Table 3), implying that photochemical aging process at noon is very efficient in transiting particles from fractal to compact morphology. It should be pointed out that there is no $D_f$ data during 12:00-17:00 under the more polluted condition, likely due to the transition of the sub-density mode particles to the main-density mode associated with active aging processes. Given that $D_f$ of the more polluted period is consistently 0.20 higher than the less polluted condition (Fig. S16) and aerosol aging process at noon is very active, the $D_f$ increase rate from 11:00 to 13:00 under the more polluted condition is calculated by assuming a $D_f$ of 3.0 at 13:00.

### 3.5 Influence of chemical composition on effective density

Chemical composition is one of the key factors affecting particle density. Inorganic components including $NH_4NO_3$ and $(NH_4)_2SO_4$ have higher density than OA and BC. Therefore, the variation in aerosol chemical composition can, to some extent, explain the wide range of observed effective density. Since 300 nm is the closest one to the peak position of particle volume distribution (Fig. S17) among all the five measured sizes, the average effective density of particles at 300 nm ($\bar{\rho}_{eff,tot,300nm}$) is used to compare with the ACSM-derived bulk effective density ($\rho_{eff,ACSM}$) based on Eq. 3 to understand the connection between particle effective density and chemical composition.

As mentioned earlier, there exists large variation in BC effective density due to the influence of morphology, but most studies have ignored the influence of BC effective density or just used a material density of 1.80 g cm$^{-3}$ for BC (Lee et al., 2013; Levy et al., 2013; Lin et al., 2018). And assuming a dense spherical BC with density of 1.80 g cm$^{-3}$ may even lead to ~17% overestimation of absorption enhancement (Zhang et al., 2016b). To analysis the influence of BC effective density on $\rho_{eff,ACSM}$, a sensitivity test with BC effective density ranging from 0.30 to 1.80 g cm$^{-3}$ and a step size of 0.05 g cm$^{-3}$ is performed in this study. The minimum value (0.30 g cm$^{-3}$) is the lowest effective density of BC at 300 nm observed in the atmosphere (Olfert et al., 2017) while the maximum (1.80 g cm$^{-3}$) is the material density of BC. The mean square error (MSE) and R-square ($R^2$) between $\bar{\rho}_{eff,tot,300nm}$ and $\rho_{eff,ACSM}$ at each BC density step are shown in Fig. 7a. In addition, Fig. 7b shows the comparison between $\bar{\rho}_{eff,tot,300nm}$ and $\rho_{eff,ACSM}$ with assumed BC effective density of 0.30, 0.60, 1.00 and 1.80 g cm$^{-3}$, respectively.

We find that $\rho_{eff,ACSM}$ can be decreased on average by 40% if the applied BC effective density changes from 1.80 to 0.30 g cm$^{-3}$. In addition, there is 34% of observed $\bar{\rho}_{eff,tot,300nm}$ with values below 1.20 g cm$^{-3}$, which could only be explained by low BC effective density. Therefore, applying a proper BC effective density is crucial for accurate estimates of the effective density of ambient particles. Among all tested values, $\rho_{eff,ACSM}$ with assumed BC effective density of 0.60 g cm$^{-3}$ matches best with measured $\bar{\rho}_{eff,tot,300nm}$ ($R^2 = 0.62$ and MSE = 0.014). Well correlation with $R^2$ of 0.62 (0.61) is found for BC effective density of 0.30 g cm$^{-3}$ (or 1.00 g cm$^{-3}$), but the MSE reaches up to 0.088 (0.048). In contrast, when using a BC effective density of 1.80 g cm$^{-3}$, the derived $\rho_{eff,ACSM}$ is much higher than the measured $\bar{\rho}_{eff,tot,300nm}$ and there is barely any correlation between them. Since strong diurnal variation was observed for BC mass fraction (Fig. S14) driven by the changes in primary source

emissions, we also conduct a sensitivity analysis to retrieve BC effective density for every 3-hour of the day. The retrieved BC effective density indeed shows a diurnal pattern (Fig. S18), with high values in the afternoon and night and relatively low values during the morning and evening, which matches well with the diurnal pattern of $F_{sub}$ (Fig. 5). However, the range of the diurnal variation (0.52-0.64 g cm$^{-3}$) is relatively small. When applying this diurnal pattern of BC effective density in the calculation of $\rho_{eff,ACSM}$, only a marginal increase was found in the $R^2$ of the correlation between $\bar{\rho}_{eff,tot,300nm}$ and $\rho_{eff,ACSM}$ ($R^2$ increased from 0.62 to 0.65, Fig. S19), probably due to the limited amount of data in each time interval and the use of bulk chemical composition in the calculation. Therefore, a fixed BC effective density of 0.60 g cm$^{-3}$ is used for the $\rho_{eff,ACSM}$ calculation in the following analysis.

Additionally, a sensitivity test for OA density ranging from 1.20 to 1.60 g cm$^{-3}$ (Dinar et al., 2006; Kostenidou et al., 2007; Turpin and Lim, 2001) is conducted as well to evaluate the uncertainties associated with the variability of OA density. As seen in Fig. S20, the changes in OA density have little influence on $\rho_{eff,ACSM}$ (difference within 8%) for the test OA density range. The MSE associated with OA densities is an order of magnitude lower than that with BC densities, and reaches minimum for OA density of ~1.20-1.30 g cm$^{-3}$. The $R^2$ at each OA density maintains around 0.62. The results suggest insignificant influence associated with the variation of OA density, in contrast to BC, and the selection of OA density of 1.30 g cm$^{-3}$ is appropriate for the effective density calculation of ambient particles.

As shown in Fig. 7b, while the majority of data points are clustered near the 1:1 line, there exists some anomalous points deviated from the 1:1 line. Appling the same PM$_{0.7}$ threshold in Sect. 3.3 (PM$_{0.7}$ volume concentration of 50 μm$^3$ cm$^{-3}$), the data sets can be separated into two groups: (i) a consistent pair with a slope of 1.16 and $R^2$ of 0.69 for the less polluted condition, and (ii) uncorrelated set with $R^2$ of 0.19 for the more polluted condition (Fig. 8). This implies that $\bar{\rho}_{eff,tot,300nm}$ could be used as a proxy for bulk effective density under the less polluted condition but not the more polluted condition. A correlation analysis is then conducted to further evaluate the influence of chemical composition on the $\bar{\rho}_{eff,tot,300nm}$ under the less polluted condition. As shown in Fig. 9a, there is an obvious positive correlation ($R^2 = 0.58$ and slope = 0.01) between $\bar{\rho}_{eff,tot,300nm}$ and SIA mass fraction, indicating that particle densities are largely dependent on SIA, consistent with Yin's and Zhang's study (Yin et al., 2015; Zhang et al., 2016a). On the contrary, a notable negative correlation is observed between $\bar{\rho}_{eff,tot,300nm}$ and BC mass fraction with $R^2$ of 0.68 and slope of -0.02 (Fig. 9b). The magnitude of both the correlation coefficient and the slope for BC are higher than SIA, implying a stronger influence of BC on the effective density compared with SIA. This further confirms the importance and necessity of including BC in the estimate of effective density for ambient particles. In addition, the correlation is poor between $\bar{\rho}_{eff,tot,300nm}$ and other chemical components.

To better indicate the variation of effective density with particle chemical composition, a sensitivity test is conducted based on arbitrarily assumed mass fractions for each component. Considering the major contribution of NH$_4$NO$_3$ in SIA, only one density of 1.73 g cm$^{-3}$ is used for total SIA, and the test is thus performed with a 3-component (SIA, BC and OA) parameterization. The results are presented in Fig.10, with the fractions of BC and SIA labelled on the x and y-axis, respectively, whereas the fractions of OA could be deduced as the rest. Calculated $\rho_{eff}$ spreads from 0.60 to 1.73 g cm$^{-3}$, corresponding to

450 100% of BC and SIA, respectively, similar to the measured range of 0.64-1.64 g cm$^{-3}$ for $\bar{\rho}_{eff,tot,300nm}$. In general, $\rho_{eff}$ increases as the BC fraction decreases or SIA fraction increases. It is interesting to note that $\rho_{eff}$ is more sensitive to BC as the fraction of BC decreases. In addition, when BC fraction < 30%, which is generally observed in the ambient (Ding et al., 2016; Yang et al., 2011), $\rho_{eff}$ is much more sensitive to the changes in BC fractions rather than SIA fractions, consistent with the results based on the correlation analysis under the less polluted condition.

In the case of the more polluted condition, $\bar{\rho}_{eff,tot,300nm}$ is concentrated close to 1.20 g cm$^{-3}$ with a standard deviation of 0.04, while $\rho_{eff,ACSM}$ varies in the range of 1.17-1.47 g cm$^{-3}$ (~0-22% higher than $\bar{\rho}_{eff,tot,300nm}$). The difference can be attributed to the discrepancies of chemical composition between bulk PM$_1$ and particles with size of 300 nm. Compared with the less polluted condition, an increase of 12.8% in SIA mass fraction (dominated by NH$_4$NO$_3$) and a decrease of 4.6% in BC mass fraction in bulk PM$_1$ is observed during the more polluted periods (Fig. S21). The mass concentration of SIA increased

from 16 to 50 µg m$^{-3}$ and most of the additional mass might be added to particles with size larger than 300 nm (Yan et al., 2021). This could be indicated by the change in particle volume size distribution (Fig. S17), which shows a decrease of volume fraction for particles with size < 300 nm in PM$_{0.7}$ from 43% at the less polluted condition to 29% at the more polluted condition. This shift towards larger particles under the more polluted condition makes the difference in chemical composition between PM$_1$ and particles at size of 300 nm noticeable and thus results in the inconsistence between $\bar{\rho}_{eff,tot,300nm}$ and $\rho_{eff,ACSM}$.

Since NH$_4$NO$_3$ and BC exhibit the largest difference between the more polluted and the less polluted conditions (Fig. S21), a sensitivity study with scale factors ranging from 0.01 to 5 applied for the two components is performed for the more polluted condition to help explain the discrepancy between $\bar{\rho}_{eff,tot,300nm}$ and $\rho_{eff,ACSM}$ (Fig. 8a). Figure 11 shows the ratios of new calculated $\rho_{eff,ACSM}$ versus $\bar{\rho}_{eff,tot,300nm}$ as well as the new fractions of NH$_4$NO$_3$ and BC in bulk PM$_1$ obtained with different scale factors. The original mean fractions of NH$_4$NO$_3$ and BC in bulk PM$_1$ under the more polluted condition are 56.3% and

11.2%, respectively (shown as red circles in Fig. 11). Assuming the increase in NH$_4$NO$_3$ occurs mainly in particles with size larger than 300 nm, the "true" fraction of NH$_4$NO$_3$ in particles with size of 300 nm would decrease while the one of BC would increase (a shift toward top-left of the plots from the original position in Fig. 11). Consequently, the discrepancy between $\rho_{eff,ACSM}$ and $\bar{\rho}_{eff,tot,300nm}$ would be reduced, with the ratios of new derived $\rho_{eff,ACSM}$ versus $\bar{\rho}_{eff,tot,300nm}$ close to 1.

## 4 Conclusion

Effective density, as one of the most important physical properties of atmospheric aerosol particles, is highly related with particle morphology and chemical composition. Here we report size-resolved particle effective densities observed during McFAN in autumn 2019 at a rural site in the North China Plain and their evolution associated with emissions and aging processes.

With a new developed flexible Gaussian fit algorithm, frequent bimodal distribution of particle effective density including 

480 a sub-density mode and a main-density mode is identified, accounting for 77-87% of total observations. The number fraction

of the sub-density mode particles ($F_{sub}$) is 22-27% for particle size from 50 to 300 nm. The prevalence of the sub-density mode is mainly related to freshly emitted black carbon (BC). Opposite size-dependent trends are found for the geometric mean of the main- and sub-density mode ($\bar{\rho}_{eff,main}$ and $\bar{\rho}_{eff,sub}$). $\bar{\rho}_{eff,main}$ increases from 1.18 g cm$^{-3}$ (50 nm) to 1.37 g cm$^{-3}$ (300 nm) attributed to larger fraction of high-density components and more significant restructuring effect at large particle sizes, while $\bar{\rho}_{eff,sub}$ decreases from 0.89 g cm$^{-3}$ (50 nm) to 0.62 g cm$^{-3}$ (300 nm) mainly ascribed to the agglomerate effect of BC. There also exist obvious diurnal cycles for both $\bar{\rho}_{eff,main}$ and $\bar{\rho}_{eff,sub}$, with maximum in the early afternoon and a relatively low level in the night. The high effective density in the afternoon could be attributed to the increased mass fraction of high material density components associated with secondary aerosol production. Both meteorological conditions and pollution levels may affect particle effective density. Air masses with sufficient aging process leading to higher $\bar{\rho}_{eff,main}$ and $\bar{\rho}_{eff,sub}$, and lower $F_{sub}$. $\bar{\rho}_{eff,main}$ in the more polluted periods is lower than that in the less polluted periods, while the difference of $\bar{\rho}_{eff,sub}$ is insignificant.

To investigate the impact of chemical composition, the average effective density of particles at 300 nm ($\bar{\rho}_{eff,tot,300nm}$) is compared with the ACSM-derived bulk effective density ($\rho_{eff,ACSM}$). The best agreement between $\bar{\rho}_{eff,tot,300nm}$ and $\rho_{eff,ACSM}$ is achieved with an assumed BC effective density of 0.6 g cm$^{-3}$. The comparison between $\rho_{eff,ACSM}$ and $\bar{\rho}_{eff,tot,300nm}$ is different for the more polluted and the less polluted conditions. Under the less polluted conditions, $\bar{\rho}_{eff,tot,300nm}$ is well correlated with $\rho_{eff,ACSM}$ with a slope close to 1, while $\bar{\rho}_{eff,tot,300nm}$ concentrated at 1.20 g cm$^{-3}$ but $\rho_{eff,ACSM}$ varies from 1.17 to 1.47 g cm$^{-3}$ under the more polluted conditions. The poor comparison under the more polluted conditions is likely to stem from the relatively larger discrepancy of chemical composition in bulk and 300 nm particles. Based on the correlation between $\bar{\rho}_{eff,tot,300nm}$ and individual particle chemical components, the mass fractions of secondary inorganic aerosol (SIA) and BC are found to be the major factors determining particle effective densities. Moreover, the influence of BC on the effective density is even stronger than that of SIA, implying the importance of including BC in the estimate of ambient particle effective densities.

**Data availability**

The data used in this study are available from the corresponding author upon reasonable request.

**Author contribution**

NM and QW conceived this research. YC, HS, QZ, PF, NM, and QW planned the McFAN campaign. YZ, CC, JT, JH, LX and SZ conducted the field measurement. YZ performed the data analysis. The written article was prepared by YZ, NM, QW and ZW with input from all other co-authors.

## Competing interests

The authors declare that they have no conflict of interest.

## Acknowledgements

This work was supported by the National Natural Science Foundation of China (41877303 and 41907182), the Foundation of State Key Laboratory of Loess and Quaternary Geology, Institute of Earth Environment, CAS (SKLLOG2029), the Guangdong Innovative and Entrepreneurial Research Team Program (Research team on atmospheric environmental roles and effects of carbonaceous species: 2016ZT06N263), and Special Fund Project for Science and Technology Innovation Strategy of Guangdong Province (2019B121205004).

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

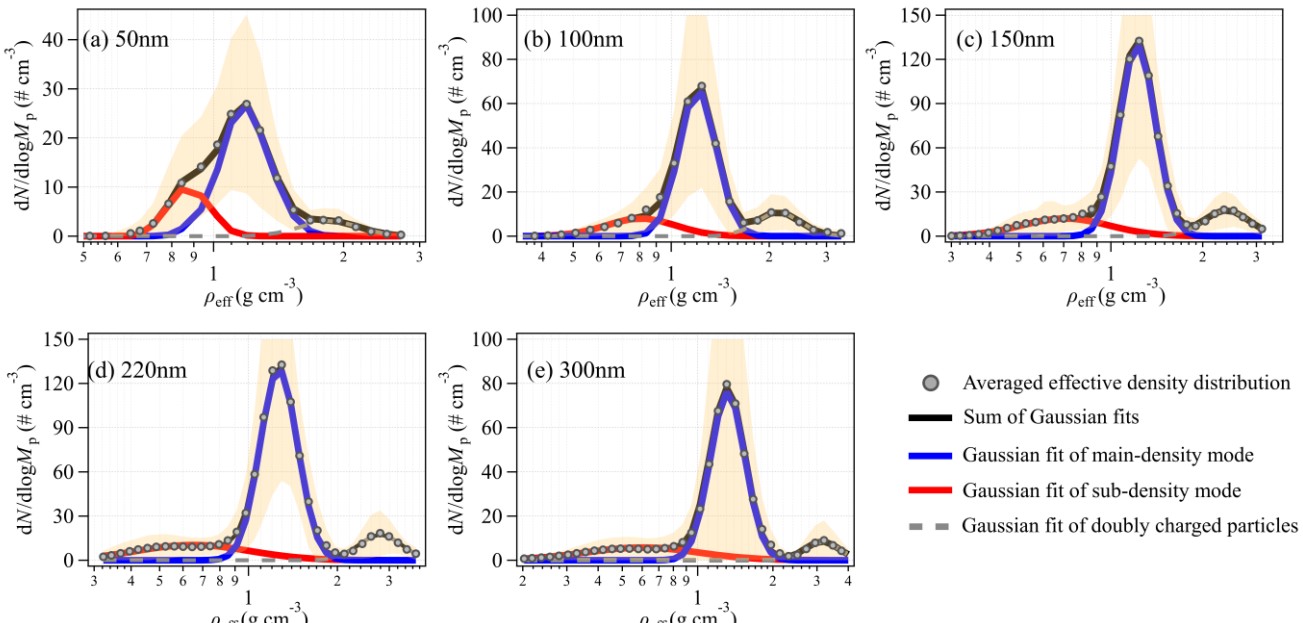

**Figure 1: Averaged particle effective density ($\rho_{eff}$) distribution for (a) 50 nm, (b) 100 nm, (c) 150 nm, (d) 220 nm, and (e) 300 nm particles during the entire sampling period. The grey points represent the measured average effective density distribution, and yellow shadowed area represents the standard deviation of the averaged data. The blue and red line are Gaussian fit of the main-density and sub-density mode at each size, respectively, and the grey dashed line represents Gaussian fit of doubly charged particles. The black line is the sum of the main-density, sub-density mode and doubly charged particles.**

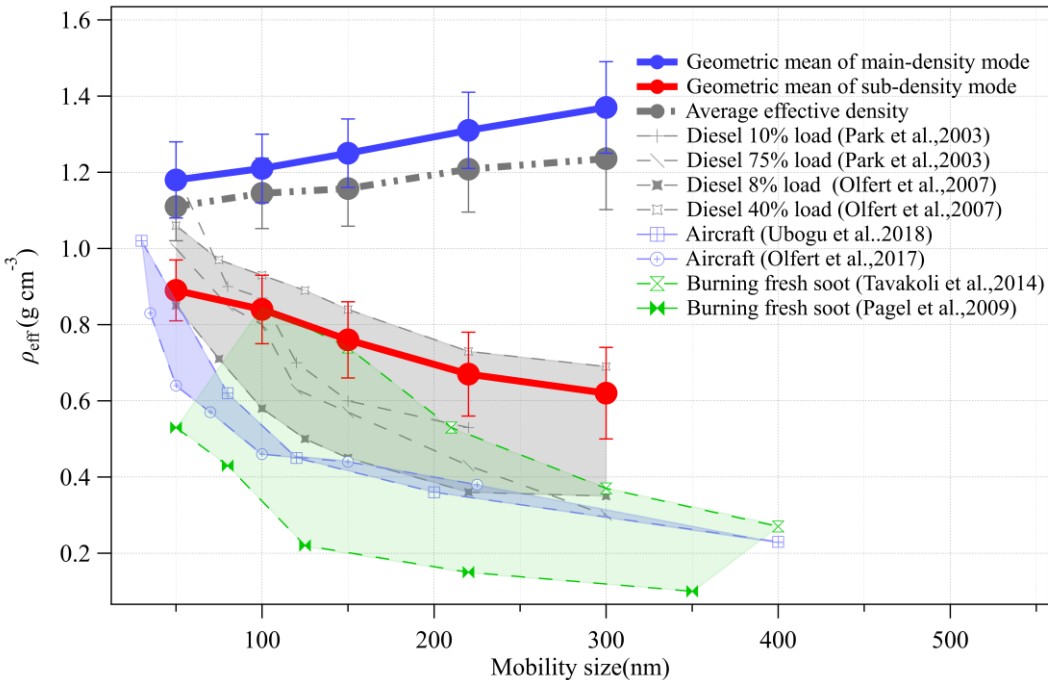

**Figure 2: Size dependency of effective density ($\rho_{eff}$) in this study and from selected laboratory experiments for comparison.**

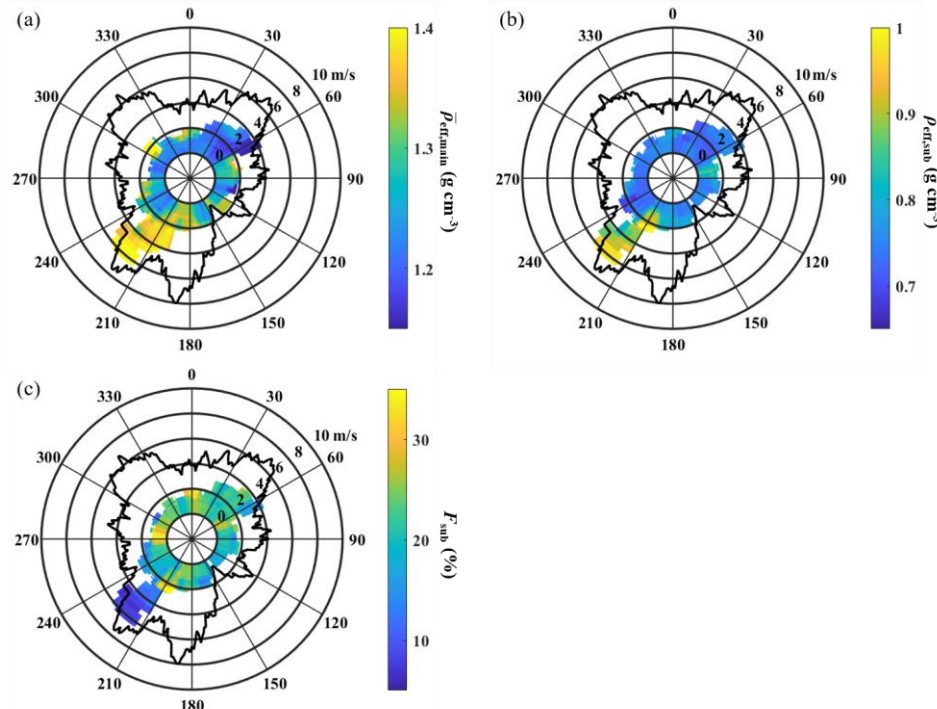

**Figure 3: Wind rose analysis of (a) geometric mean of the main-density mode ($\bar{\rho}_{\text{eff,main}}$), (b) geometric mean of the sub-density mode ($\bar{\rho}_{\text{eff,sub}}$), (c) number fraction of the sub-density mode ($F_{\text{sub}}$) for 150 nm particles. Black bold lines represent wind frequency during the entire sampling period.**


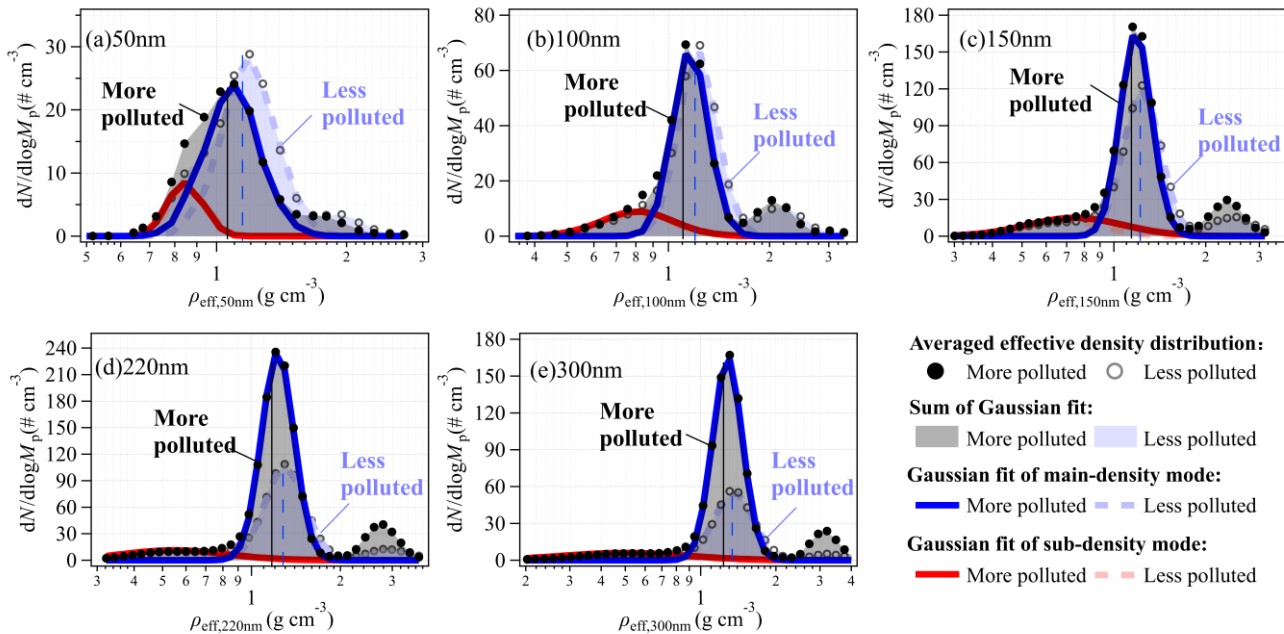

**Figure 4: Averaged particle effective density (ρ$_{eff}$) distribution for (a) 50 nm, (b) 100 nm, (c) 150 nm, (d) 220 nm, and (e) 300 nm particles under the more and the less polluted conditions.**

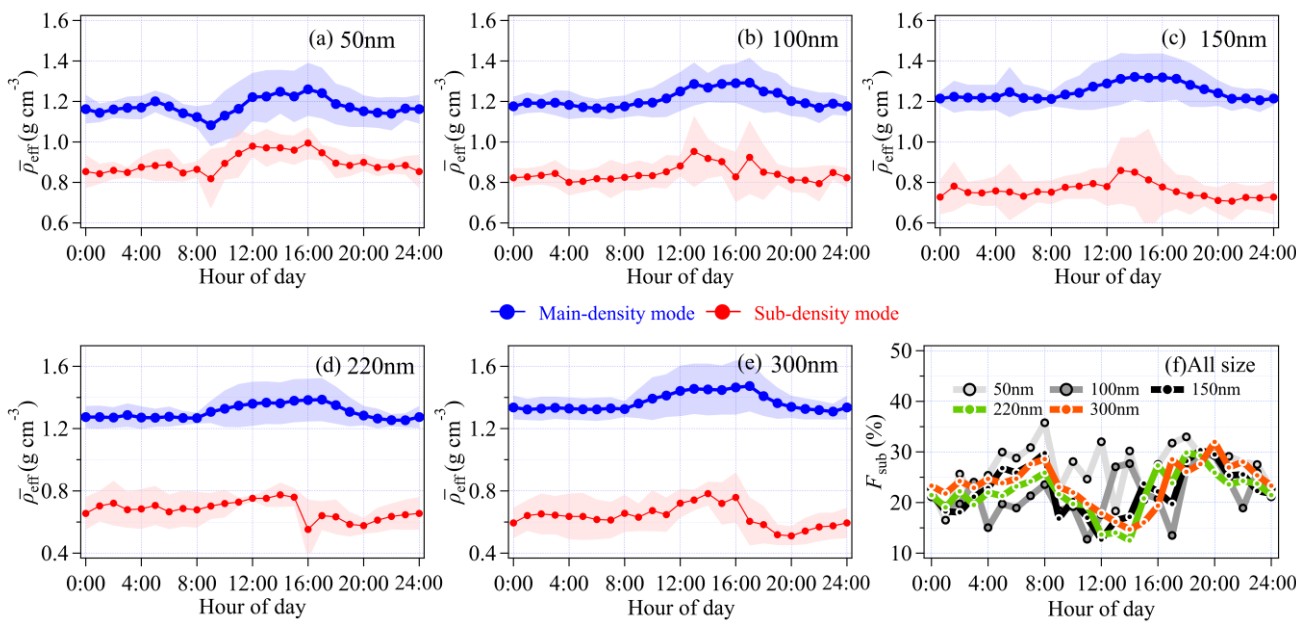

**Figure 5: Diurnal cycle of: geometric mean effective density ($\bar{\rho}_{\mathrm{eff}}$) of the main-density and sub-density mode for (a) 50 nm, (b) 100 nm, (c) 150 nm, (d) 220 nm, and (e) 300 nm particles; (f) number fraction of the sub-density mode ($F_{\mathrm{sub}}$).**


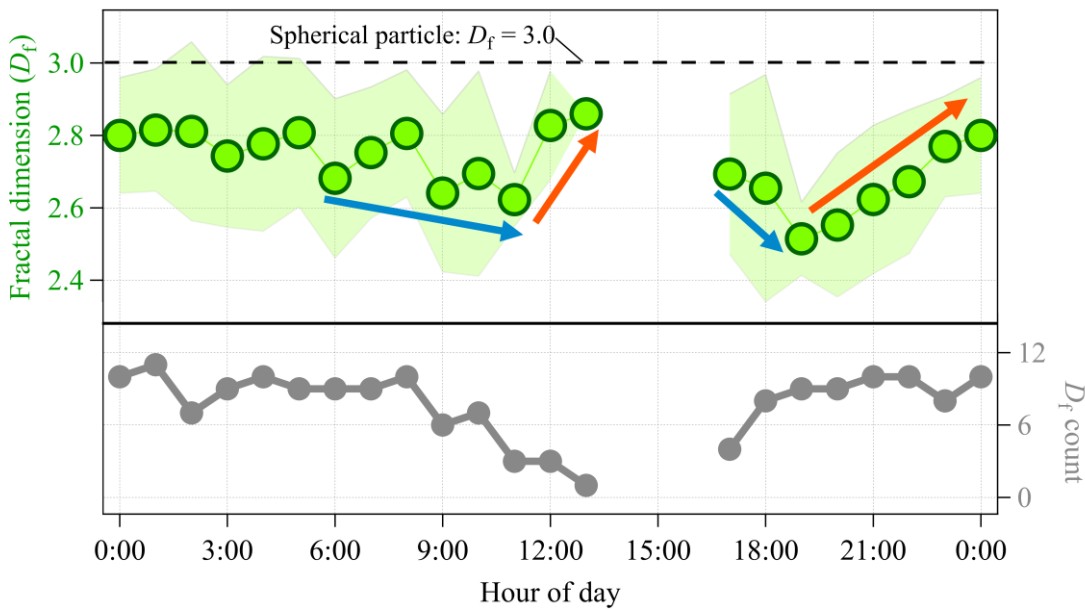

**Figure 6: Diurnal cycle of fractal dimension ($D_f$) of the sub-density mode particles. The dotted line is $D_f$ with the value of 3.0, indicating particle with a spherical morphology. The blue arrows represent $D_f$ with decreasing trend and the orange arrows represent $D_f$ with increasing trend.**

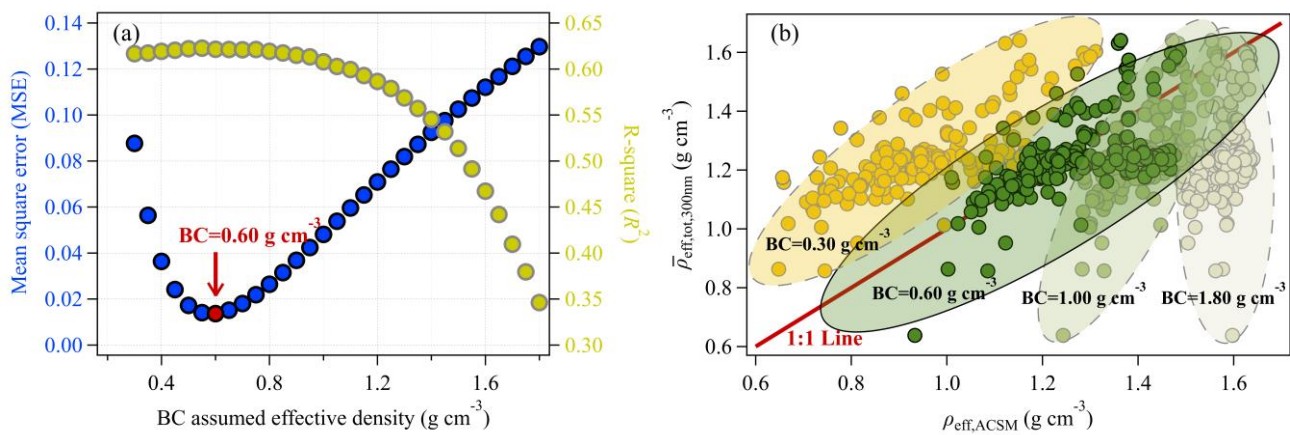


**Figure 7: (a)** Mean square error (MSE) and R-square ($R^2$) results from BC effective density sensitivity test. **(b)** Comparison of the average effective density of particles at 300 nm observed by DMA-CPMA-CPC ($\bar{\rho}_{eff,tot,300nm}$) and ACSM-derived bulk effective density ($\rho_{eff,ACSM}$). The color shaded areas represent $\rho_{eff,ACSM}$ based on different BC effective density assumption. Red line is the line with slope of 1.

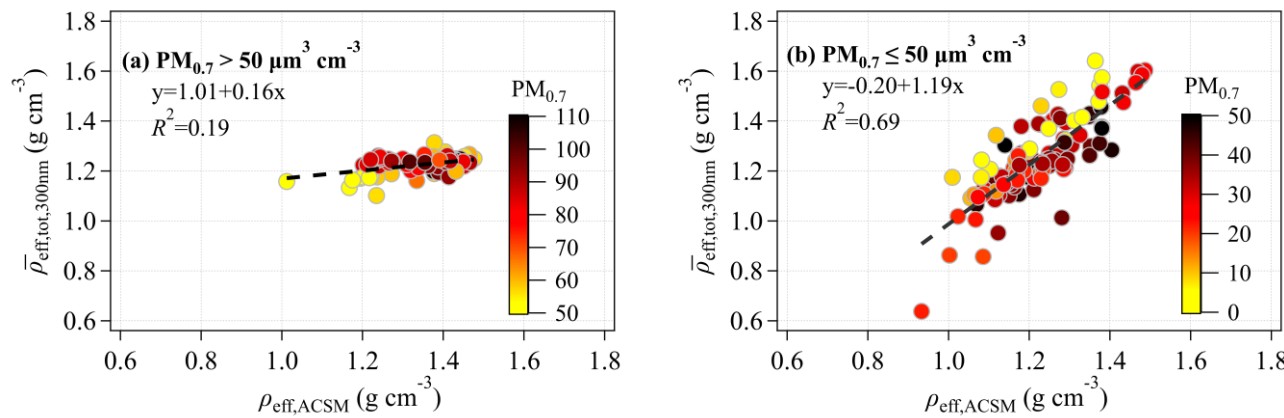


**Figure 8: Comparison of the average effective density of particles at 300 nm observed by DMA-CPMA-CPC ($\overline{\rho}_{\text{eff,tot,300nm}}$) and ACSM-derived bulk effective density ($\rho_{\text{eff,ACSM}}$) under (a) the more polluted (corresponding to PM$_{0.7}$ > 50 µm$^3$ cm$^{-3}$) and (b) the less polluted (corresponding to PM$_{0.7}$ ≤ 50 µm$^3$ cm$^{-3}$) condition. Colored circles represent PM$_{0.7}$ volume concentrations.**

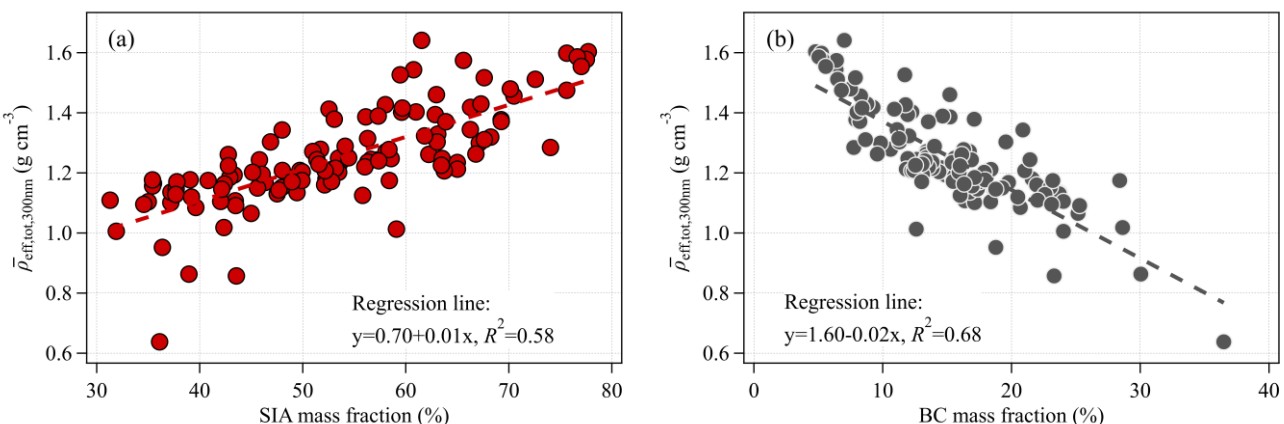

**Figure 9: Correlation between (a) SIA, (b) BC mass fraction and $\overline{\rho}_{eff,tot,300nm}$ under the less polluted condition.**

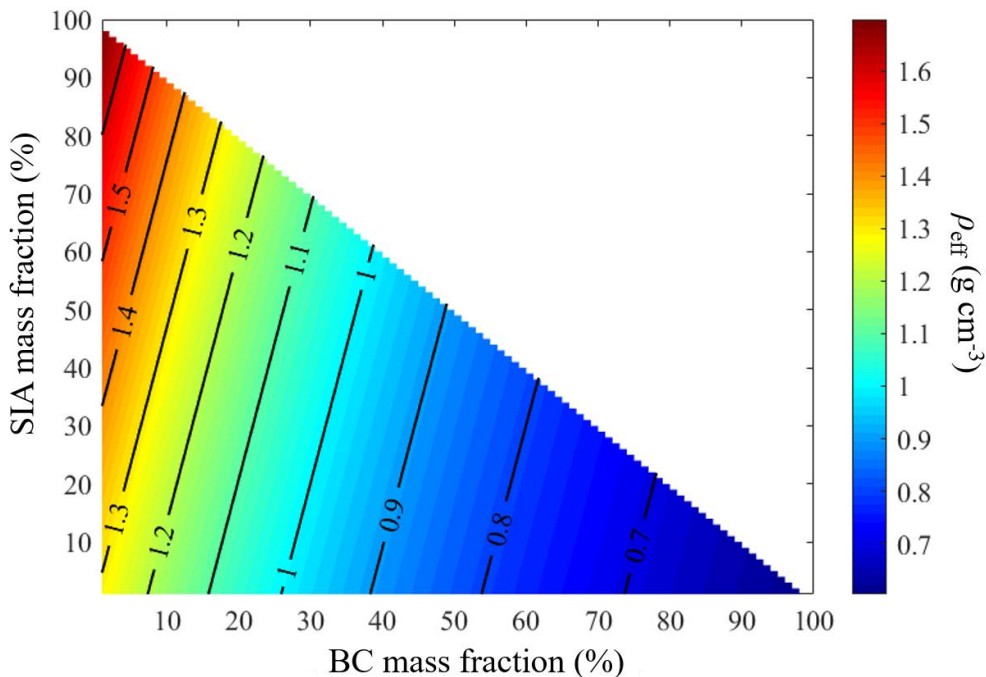

**Figure 10: Particle effective density ($\rho_{eff}$) calculation based on assumed chemical composition.**

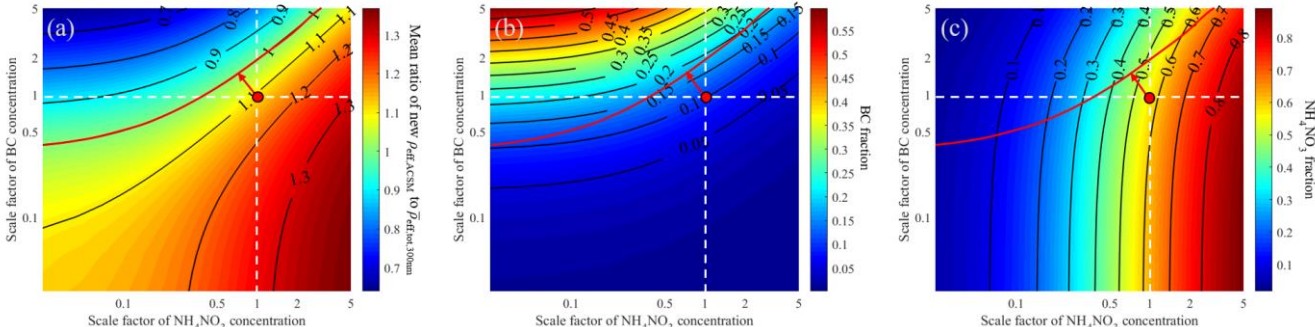

**Figure 11: (a)** Mean ratio of new calculated $\rho_{eff,ACSM}$ versus measured $\overline{\rho}_{eff,tot,300nm}$ based on scale factor of NH₄NO₃ and BC mass concentrations ranging from 0.01 to 5 (at logarithmic scale), **(b)** the corresponding NH₄NO₃ mass fraction and **(c)** the corresponding BC mass fraction under polluted condition. The black lines represent contours with the same values. The vertical and horizonal white dotted lines respectively represent NH₄NO₃ and BC mass concentration at the scale factor of 1, and red circles are original mass concentrations of BC and NH₄NO₃.

**Table 1: The geometric mean effective density ($\bar{\rho}_{\mathrm{eff,sub}}$ and $\bar{\rho}_{\mathrm{eff,main}}$) and standard deviation ($\sigma_{\mathrm{sub}}$ and $\sigma_{\mathrm{main}}$) of the sub-density and main-density mode, number fraction ($F_{\mathrm{sub}}$) and occurrence frequency of the sub-density mode during the entire sampling period.**

| Size(nm) | $\bar{\rho}_{\mathrm{eff,sub}}$(g cm$^{-3}$) | $\bar{\rho}_{\mathrm{eff,main}}$(g cm$^{-3}$) | $\sigma_{\mathrm{main}}$ | $\sigma_{\mathrm{sub}}$ | $F_{\mathrm{sub}}$(%) | Frequency(%) |
|---|---|---|---|---|---|---|
| 50 | 0.89±0.08 | 1.18±0.10 | 0.053±0.012 | 0.044±0.014 | 27±18 | 79 |
| 100 | 0.84±0.09 | 1.21±0.09 | 0.051±0.010 | 0.074±0.027 | 22±11 | 87 |
| 150 | 0.76±0.10 | 1.25±0.09 | 0.053±0.007 | 0.129±0.034 | 23±12 | 86 |
| 220 | 0.67±0.11 | 1.31±0.10 | 0.059±0.008 | 0.172±0.037 | 23±10 | 77 |
| 300 | 0.62±0.12 | 1.37±0.12 | 0.067±0.010 | 0.215±0.036 | 25±10 | 81 |

**Table 2: The geometric mean effective density ($\bar{\rho}_{\mathrm{eff,sub}}$ and $\bar{\rho}_{\mathrm{eff,main}}$) of the sub-density and main-density mode, number fraction ($F_{\mathrm{sub}}$) and occurrence frequency of the sub-density mode under the more and the less polluted conditions.**

| Size(nm) | Condition | $\bar{\rho}_{\mathrm{eff,sub}}$(g cm$^{-3}$) | $\bar{\rho}_{\mathrm{eff,main}}$(g cm$^{-3}$) | $F_{\mathrm{sub}}$(%) | Frequency(%) |
|---|---|---|---|---|---|
| 50 | More polluted | 0.87±0.06 | 1.11±0.09 | 42±25 | 72 |
| | Less polluted | 0.90±0.08 | 1.19±0.10 | 24±13 | 81 |
| 100 | More polluted | 0.83±0.07 | 1.15±0.05 | 25±11 | 95 |
| | Less polluted | 0.84±0.09 | 1.23±0.09 | 21±11 | 84 |
| 150 | More polluted | 0.74±0.05 | 1.18±0.04 | 20±8 | 89 |
| | Less polluted | 0.76±0.10 | 1.27±0.09 | 24±12 | 86 |
| 220 | More polluted | 0.68±0.10 | 1.23±0.03 | 18±3 | 63 |
| | Less polluted | 0.67±0.11 | 1.33±0.11 | 24±11 | 82 |
| 300 | More polluted | 0.65±0.13 | 1.29±0.03 | 18±4 | 57 |
| | Less polluted | 0.62±0.12 | 1.40±0.12 | 26±11 | 89 |

**Table 3: Comparison of fractal dimension ($D_f$) of the sub-density mode particles at noon and at night under the more and the less polluted conditions.**

| | $D_f$ at 11:00 | $D_f$ at 13:00 | Increase rate ($h^{-1}$) | $D_f$ at 19:00 | $D_f$ at 24:00 | Increase rate ($h^{-1}$) |
|---|---|---|---|---|---|---|
| All | 2.62 | 2.86 | 0.12 | 2.51 | 2.80 | 0.06 |
| More polluted | 2.71 | — | 0.145* | 2.56 | 2.98 | 0.08 |
| Less polluted | 2.58 | 2.86 | 0.14 | 2.49 | 2.75 | 0.05 |

\* is calculated by assuming $D_f$ at 13:00 is 3.0.