# Peer review of "Bimodal distribution of size-resolved particle effective density: results from a short campaign in a rural environment in the North China Plain"

_Atmospheric Chemistry and Physics, 2021_

## Author Comment (AC1)

**Response to reviewer #1**

We thank the reviewer for those supportive and thoughtful comments. Our responses to the comments are provided below in blue, with the reviewers' comments in black.

**General Comments:**

In this study, a combined DMA-CPMA-CPC system was applied to characterize the size-resolved particle effective density in Multiphase chemistry experiment in Fogs and Aerosols in the North China Plain (McFAN) in autumn 2019. They identified a frequent bimodal distribution of particle effective density, and a unique low-density mode (named sub-density mode) accounted for ~20-30% of total observed particles. The diurnal variations of particle effective density and the influence of pollution and secondary aerosols were discussed. They concluded that the influence of BC on the effective density is even stronger than SIA.

Overall, the paper is well-written and is appropriate for ACP. The results clearly indicate the factors that govern the variations of particle effective density. The size-resolved particle effective density shown in the manuscript is interesting and would have implications for further studies. Some minor comments are still needed to be addressed before the manuscript can be published.

Response:

Thank you for the positive feedback and helpful suggestions. We have addressed the comments and implemented all suggestions in the revised manuscript as detailed below.

**Specific Comments**

1. The authors directly linked the sub-density mode to fresh black carbon (BC) emissions. Some organics might also have very low densities, which might lead to ambiguous conclusions. Previous measurements have also indicated that organics dominated in smaller size ranges. This is a key requirement when clarify the significance of BC in such mode.

Response:

Thanks for the comment. We agree with the reviewer that some organics might have very low densities and contribute to the sub-density mode. Therefore, we tried to assess the possible contribution of BC and organics on the sub-density mode via the correlation analysis between the number fraction of the sub-density mode ( $F_{sub}$ ) and the mass fractions of BC and organics in PM1. Unfortunately, there was no measurement of the chemical composition of ultrafine particles (diameter < 100 nm) during our experiment and we could not determine the actual contribution of organics in the sub-density mode at 50 and 100 nm. Accordingly, we have added this discussion in Sect. 3.1 as: "On the other hand, a low effective density mode (density < 1.20 g cm-3) exists or even dominates in the measurements near emission sources, which is ascribed to freshly emitted particles and non-uniformly mixed particles (Nosko and Olofsson, 2017; Olfert and Rogak, 2019; Park et al., 2003). Numerous studies have found low effective densities of freshly emitted BC, with a minimum of 0.10 g cm-3 (Pagels et al., 2009). While the density of OA is usually assumed as 1.2-1.3 g cm-3 in most cases (Hallquist et al., 2009), some studies have found it could be as low as 0.6-1.1 g cm-3 (Nakao et al.,

2011; Li et al., 2016). To eluciate the role of these two components in the sub-density mode, we further analyze the correlation between the number fraction of the sub-density mode ( $F_{sub}$ ) and the mass fractions of BC and OA. As seen in Fig. S6-S7, the mass fraction of BC shows significant correlation with  $F_{sub}$  at 150, 220 and 300 nm ( $R^2 =$ 0.46-0.57), whereas barely no correlation is observed between OA and  $F_{sub}$  ( $R^2 = 0.02$ -0.09), implying that the sub-density mode at these three sizes could be mainly attributed to freshly emitted BC and the quantity of the sub-density mode is closely related to the variation of BC mass fraction. However,  $F_{sub}$  at 50-100 nm shows little correlation with either BC or OA mass fraction, which could be explained by the difference between the PM1 bulk chemical composition and the chemical composition of particles smalller than 100 nm."

Figure R1 (S6). Correlation between BC mass fraction and the number fraction of the subdensity mode ( $F_{sub}$ ) for (a) 50 nm, (b) 100 nm, (c) 150 nm, (d) 220 nm, and (e) 300 nm particles.

---

## Author Comment (AC2)

**Response to reviewer #2**

We thank the reviewer for those supportive and thoughtful comments. Our responses to the comments are provided below in blue, with the reviewers' comments in black.

Dear Zhou et al.,

thank you for the interesting study regarding effective densities of ambient aerosol particles. The manuscript "Bimodal distribution of size-resolved particle effective density in a rural environment in the North China Plain" has been written very well and it is based on experiments conducted with state-of-the-art methods. The study presented in the manuscript aims to describe the effective density of ambient particles but also link it to the sources of particles, especially in case of observation of low effective densities. The figures of the manuscript are clear and mostly very informative and tables serve very well the structure of the manuscript.

Response:

We appreciated referee#2's positive feedback and constructive suggestions which are of great value for improving the quality of our paper. Our point-to-point replies to the referee's comments are listed below.

1. One relatively important issue in the presented study is the duration of the measurement campaign. I think the experimental period is not long enough to generalize the results. Regarding to that, is it possible to modify the title and abstract so that this is brought out to readers already in the beginning of the paper? Mentioning that the study is "case study" or "short campaign" would be enough for that purpose. Response:

Thanks for the comment. We have changed the title to "Bimodal distribution of sizeresolved particle effective density: results from a short campaign in a rural environment in the North China Plain" in the revised manuscript. We also revised the abstract accordingly: "In this study, size-resolved particle effective density was measured with a combined DMA-CPMA-CPC system in autumn 2019 as part of the Multiphase chemistry experiment in Fogs and Aerosols in the North China Plain (McFAN)."

2. As we all know, weather conditions have crucial role in aerosol formation, emission transportation and emission ageing, and they affect the ambient concentrations significantly. I propose that the authors include much more detailed weather data to the paper and investigate how the weather affect the effective densities of the particles. I think the affiliations of the authors enable the access to local weather data if it was not measured directly at particle measurement site. In addition, inclusion of the weather data into the paper enables better comparisons to other studies made later in same place or in other places by other researchers.

Response:

Thanks for this very constructive suggestion. We have now included the weather data (including wind speed, wind direction, relative humidity and temperature) measured at a standard weather station  $\sim 200$  m away from the measurement container in the revised

manuscript. The overall weather condition is added to the timeseries plot (Fig. R1, added as Fig. S2 in SI). Accordingly, we have added one additional section (Sect. 3.2) as well as Fig. R2 (Fig. 3 in the revised manuscript) and Fig. R3-R4 (Fig. S10-S11 in SI) to specifically discuss the influence of meteorological conditions on both the pollution level and particle effective density:

**"3.2 Evolutions of effective density with meteorology conditions**

Weather conditions play a crucial role in the formation, aging and emission transportation of aerosol, and may therefore affect the distribution, composition, mixing state and consequently also the effective density of ambient aerosol. As seen in Fig. S2, the pollution level at the sampling site is sensitive to the variations of wind speed and direction during the observation period. Low  $PM_{0.7}$  concentrations were usually presented with strong northerly winds while high  $PM_{0.7}$  concentrations were associated with calm winds or southwest winds.

Figure 3 and Fig. S10 shows the average  $\bar{\rho}_{eff,main}$ ,  $\bar{\rho}_{eff,sub}$  and  $F_{sub}$  at each specific wind speed and direction. Obvious high values of  $\bar{\rho}_{eff,main}$  and  $\bar{\rho}_{eff,sub}$  appear with wind direction of southwest and wind speed  $> 2 \text{ m s}^{-1}$ . This pattern clearly indicates the influence of regional transport from southern Hebei Province, an area greatly affected by emissions from industrial and residentials sources (Huang et al., 2019; Li et al., 2017). Air masses from this direction may bring pollutants with sufficient aging process, leading to changes in particle chemical composition and morphology, and consequently an increase in the fraction of particles closer to spherical with higher effective densities. Accordingly,  $F_{sub}$  also shows distinctly low values (Fig. 3c and Fig. S10). It is worth mentioning that  $\bar{\rho}_{eff,main}$  and  $\bar{\rho}_{eff,sub}$  do not show any obvious difference for wind direction of northwest. This implies that the influence of the traffic emission at No.107 National Way, which is approximate 1.5 km away from the sampling site (Fig. S1), on our measurements is somehow limited. We also noticed that an increasing trend of  $\bar{\rho}_{eff,main}$  is presented with increasing wind speed (Fig. S11). This increase could be interpreted by the antagonism between well-aged particles from long-range transport and fresh particles from local emissions. High wind speed is usually accompanied with the long-range transport of particles with sufficient aging and consequently high effective density; while low wind speed generally implies higher contribution of local fresh emissions, resulting in more particles with non-spherical morphology."

---

## Author Response (AR2)

Dear Editor,

Thank you very much for the supportive feedback and consideration of our paper for final publication on ACP. Our point-to-point replies to the technical corrections are listed below:

1. Abstract: line 25: please remove the first % mark (22-27%). Please correct the other occurrences in the text as well.

   We have removed the first '%' in the sentence. We also checked all units in the manuscript and made corrections accordingly.

2. Abstract: line 25: total number of observed particles.

   This sentence has been changed as: "… *with a low-density mode (named sub-density mode) accounting for 22-27% of total number of observed particles.*"

3. Introduction: line 95, please indicate the time and location of the campaign already here: (McFan) in Hebei province in October-November, 2019.

   We have added the time and location of the campaign as suggest: "*In this study, size-resolved effective density of ambient particles was measured with a DMA-CPMA-CPC system during Multiphase chemistry experiment in Fogs and Aerosols in the North China Plain (McFAN) in Hebei province in October-November, 2019.*"

4. Methods: line 128: Please correct the sentence starting "Particles with size ranging..." for example as "The measurements were conducted for the aerosol particles in the size range 50 to 500 nm, which are not very sensitive ..."

   Thanks for the correction. We have revised the sentence as: "*The measurements were conducted for the aerosol particles in the size range from 50 to 500 nm, which are not very sensitive to the three main loss mechanisms (i.e., diffusion loss, sedimentation loss and impaction loss) (Baron and Willeke, 2001; Von Der Weiden et al., 2009).*"

5. Methods: line 130: The particle losses...

   Thanks for the correction. We have added 'the' in the sentence.

6. Figure 2 caption: Instead of "Trend lines", I suggest formulation the caption as follows: "Size dependency of effective density (rho_eff) in this study and from selected laboratory experiments for comparison.

   Thanks for the suggestion. The caption of Fig.2 has been revised as: "*Size dependency of effective density ($\rho_{eff}$) in this study and from selected laboratory experiments for comparison.*".

We really appreciate your time invested in our paper. We would like to extend our warm wishes for the upcoming holiday season, and wish you and your family a Merry Christmas!

Best regards,
Nan and Qiaoqiao